# Aerobic and Anaerobic Bacterial and Fungal Degradation of Pyrene: Mechanism Pathway Including Biochemical Reaction and Catabolic Genes

**DOI:** 10.3390/ijms22158202

**Published:** 2021-07-30

**Authors:** Ali Mohamed Elyamine, Jie Kan, Shanshan Meng, Peng Tao, Hui Wang, Zhong Hu

**Affiliations:** 1Key Laboratory of Resources and Environmental Microbiology, Department of Biology, Shantou University, Shantou 515063, China; elyoh@hotmail.fr (A.M.E.); 13jkan@stu.edu.cn (J.K.); 17ssmeng@stu.edu.cn (S.M.); tpeng@stu.edu.cn (P.T.); wanghui@stu.edu.cn (H.W.); 2Department of Life Science, Faculty of Science and Technology, University of Comoros, Moroni 269, Comoros

**Keywords:** polycyclic aromatic hydrocarbon, pyrene, degradation mechanism pathway, bacteria, fungi, metabolism

## Abstract

Microbial biodegradation is one of the acceptable technologies to remediate and control the pollution by polycyclic aromatic hydrocarbon (PAH). Several bacteria, fungi, and cyanobacteria strains have been isolated and used for bioremediation purpose. This review paper is intended to provide key information on the various steps and actors involved in the bacterial and fungal aerobic and anaerobic degradation of pyrene, a high molecular weight PAH, including catabolic genes and enzymes, in order to expand our understanding on pyrene degradation. The aerobic degradation pathway by *Mycobacterium vanbaalenii* PRY-1 and *Mycobactetrium* sp. KMS and the anaerobic one, by the facultative bacteria anaerobe *Pseudomonas* sp. JP1 and *Klebsiella* sp. LZ6 are reviewed and presented, to describe the complete and integrated degradation mechanism pathway of pyrene. The different microbial strains with the ability to degrade pyrene are listed, and the degradation of pyrene by consortium is also discussed. The future studies on the anaerobic degradation of pyrene would be a great initiative to understand and address the degradation mechanism pathway, since, although some strains are identified to degrade pyrene in reduced or total absence of oxygen, the degradation pathway of more than 90% remains unclear and incomplete. Additionally, the present review recommends the use of the combination of various strains of anaerobic fungi and a fungi consortium and anaerobic bacteria to achieve maximum efficiency of the pyrene biodegradation mechanism.

## 1. Introduction

Among the most prevalent and persistent contaminants, polycyclic aromatic hydrocarbons (PAHs) have attracted increasing attention. They are a ubiquitous group of organic pollutants, consisting of two or more single or fused aromatics rings [1]. They are from both biological processes and by-products of incomplete combustion from sources of natural combustion (volcanic eruption or bush or forest fire) or human made sources (coal, charbroiled meats, cigarette, wood, garbage, petrol, oil, and other organic matter) [2,3,4]. If the industrial sector constitutes one of the key sources emitting more than 30% of PAHs in the ambient air [5], the compounds are however not chemically synthesized for industrial purposes. Although PAHs are subjected to different treatments, they are considered as persistent contaminants in the environment and present different toxicological characteristics [6]. PAHs are biologically active molecules that, once absorbed by organisms, under enzymatic activity are transformed, leading to the formation of epoxides and/or hydroxylated derivatives [7], which may be more toxic, by binding to fundamental biological molecules such as proteins, RNA, and DNA and causing cellular dysfunctions [8]. Up to now, 130 PAHs have been identified, but only 16 of them are listed by the United States Environmental Protection Agency (USEPA) as priority pollutants for their toxic, carcinogenic, and mutagenic properties [9]. Their adverse effects are not only observed in the different ecosystems, but even humans are not spared [10]. Thus, the control and management of PAH proliferation in environmental matrices has become a global concern that everyone should be aware of, in order to reduce the negative impacts on human health and the ecosystem, as well as their propagation in other environmental matrices.

Various physical and biochemical, environmentally sustainable, and acceptable methods have been developed together for this purpose. The decontamination techniques involve a substantial reduction in the quantity (volume and mass) of pollutants, by using appropriate methods such as bioremediation. PAHs are biologically eliminated or degraded under controlled conditions to a harmless state or to levels below the concentration limits established by regulatory authorities [7,11]. The microbial community is becoming increasingly attentive insofar as it helps in the bioremediation of contaminated areas and the degradation of the original anthropogenic pollutants. Due to the reduced impact and reliability, their involvement in bioremediation has become a technique adopted worldwide by scientists. Although the degradation of organic pollutants and especially that of PAHs was initiated decades ago, research to find the best degradation method through various approaches is still ongoing. The metabolic pathway of most low and high molecular weight PAHs is deeply studied and mastered. Although the microbial degradation of low molecular weight PAHs (LMW-PAHs) is well understood, that of the high molecular ones (HMW-PAHs) is far to be controlled. However, the metabolic pathway of pyrene, a HMW-PAH, containing four fused benzene rings, has emerged since 1988 and is presently on the way to be grasped.

The bacterial strain *Mycobacterium vanbaalenii* PYR-1 is reported as the first isolated bacteria capable of totally mineralizing pyrene by mono- and dioxygenase reactions [12]. However, due to metagenomics and novel technological tools, other microorganisms of different phylum are isolated and extensively studied to raise the curtain on the mechanism of pyrene degradation, including enzymatic and molecular actors. By combining pyrosequencing of the 16S rRNA genes and metagenomic functional analysis, a close relationship between chemical pollutants and microorganisms is revealed. Moreover, nowadays, it is even possible, by using clustering tools such as Metabat, to obtain genomic sequences of rare uncultured bacteria by analyzing special metagenomic data [13]. This present review paper attempts to provide key information on the various steps and actors involved in the bacterial and fungal aerobic and anaerobic degradation of pyrene, including catabolic genes and enzymes, to expand our understanding on pyrene degradation. This article provides a solid basis on the mechanisms of pyrene degradation as well as the actors involved.

## 2. Bacteria

Bacteria are the most numerous microorganisms, typically reaching about 10^8^ to 10^12^ cells in 1 g sample of rhizosphere soil [14] and up to 10^26^ cells in the phyllosphere [15]. As an important part of the global ecosystem, bacteria play a major role in the microbial degradation of organic pollutants [16]. Since Davies and Evans reported for the first time that Pseudomonad bacteria can degrade naphthalene, in the past few decades, significant progress in the study of biological species, degradation pathways, and catabolic genes has been made by scholars. The bacteria that have a strong ability to degrade PAHs mainly include *Rhodococcus*, *Pseudomonas*, *Sphingomonas*, *Bacillus*, *Flavobacterium*, *Mycobacterium*, *Nocardia*, *Vibrio*, *Micrococcus*, and *Acinetobacter* [17,18,19]. *Rhodococcus*, *Mycobacterium*, *Nocadiodes,* and *Terrabacteres* are capable of degrading PAHs with high molecular weight having more than three rings [20]. Metabolic pathways including the enzymes and involved catabolic genes for several Gram-negative bacteria such as *Pseudomonas*, *Sphingomonas*, *Burkholderia,* and *Comamonas* are proposed. Although many of these bacteria have been identified as low and high molecular weight PAH-degraders, information on the genetic and biochemical mechanism is still poorly understood.

### 2.1. Strains with Ability to Degrade Pyrene

Bacterial strains, namely *Rhodococcus* HCCS, *Sphingomonas* MWFG, and *Paracoccus* SPNT, isolated from mangrove rhizosphere were able to degrade phenanthrene, fluoranthene, and pyrene with a degradation rate of 90%, 40%, and 69%, respectively [21]. *Arthrobacter* sp. MAL3 and *Microbacterium* sp. MAL2, two Gram-positive bacterial strains of Actinobacteria phylum, and one Gram-negative strain *Stenotrophomonas* sp. MAL1, belonging to the Proteobacteria phylum, were isolated from the rhizosphere of maize (*Zea mays*) and Sudan grass (*Sorghum sudanensis*) grown in PAH-contaminated soils. These three bacterial strains exhibited a complete degradation of pyrene and BaP [22]. Similarly, Ortega-Calvo et al. reported that different PAHs such as naphthalene, phenanthrene, anthracene, and pyrene were used as sole carbon source by three different *Pseudomonas* strains, namely *P*. *alcaligenes* 8A, *P. stutzeri* 9A, and *P. putida* 10D, isolated in the rhizosphere soil samples originating from PAH-polluted soils [23]. Three new bacterial strains, *Bacillus thurigiensis*, *Bacillus pumilus,* and *Rhodococcus hoagii*, were isolated from the rhizosphere of *Panicum aquaticum* and characterized as potential bioremediating bacteria with a metabolism rate of 73%, 69%, 52%, and 48%, respectively, for phenanthrene, anthracene, fluoranthene, and pyrene [24]

However, although individual bacteria are able to metabolize pyrene, consortiums of microorganisms generally show more advantages for the transformation of environmental pollutants. In general, synergistic cooperation occurs as a result of metabolic deficiencies or associated metabolisms. For example, separately, the bacteria *Rhodococcus wratislaviensis* strain 9 and the microalgae *Chorella* sp. strain MM3 have been shown to be effective for pyrene mineralization. However, the mixture of these two microorganisms used, respectively, as heterotropic partner and phototropic partner was found to be more effective in degrading almost all of the pyrene in 30 days.

### 2.2. Complete and Integrated Degradation of Pyrene

Generally, the degradation of pyrene under aerobic conditions, where the enzymes oxygenase and dioxygenases are involved occurs as follow: In the first step, pyrene is hydroxylated by a dioxygenase, to form a *cis* dihydrodiol, which is then rearomatized into a diol intermediate, by a dehydrogenation reaction. The intermediate diol is then cleaved by either intra or extradiol ring cleavage dioxygenase, through either ortho- or meta-cleavage pathway, resulting in specific intermediates such as catechol, which is then converted in the intermediate tricarboxylic acid (TCA) cycle. However, the bacteria monooxygenase metabolizes pyrene in epoxide by introducing an oxygen molecule into the aromatic ring to finally form arene epoxide intermediate, which is subsequently transformed to *trans* dihydrodiol.

#### 2.2.1. Aerobic Degradation

The aerobic degradation of low and high molecular weight PAHs is intensively studied and reported by several scholars. Distinct aerobic bacteria degrade PAHs through three main steps: the ring cleavage process, side chain metabolism process, and central metabolism process. In each step, different cascade enzymatic reactions succeed each other, mainly including dioxygenase, dihydrogendiol dehydrogenase, ring-breaking dioxygenase, epoxide hydrolase, alcohol dehydrogenase, and acetaldehyde dehydrogenation enzymes and decarboxylase [2,25,26].

Several bacterial strains, such as *Mycobacterium vanbaalenii* PRY-1, *Mycobactetrium* sp. KMS, *Mycobacterium flavescens*, M. spp., Ap1, KR, 6PY1, and RJGII-135, *Pseudomonas stutzeri* P16, *Bacillus cereus* P21, and others, can completely degrade pyrene. In the following, two different bacterial strains, *Mycobacterium vanbaalenii* PRY-1 and *Mycobactetrium* sp. KMS, are used as models to describe the completed and integrated degradation mechanism pathway of pyrene.

##### Degradation of Pyrene by *Mycobacterium vanbaalenii* PYR-1

The bacterial strain *Mycobacterium vanbaalenii* PYR-1 is reported as the first isolated bacteria capable of totally mineralizing pyrene [12]. Although the metabolic pathway of pyrene by PYR-1 is extensively studied and reported, only a few studies have reported its complete and integrated degradation, involving all the enzymes, functional genes, and catabolic genes/ORFs that encode all the different steps. As a PAH compound, pyrene is degraded through three main steps: the ring cleavage process, side chain metabolism process, and central metabolism process [25,26].

Enzymatic Reactions and Degradations Genes

Molecular degradation of PAHs or their chemical structure transformation is mainly by enzymatic reactions, catalyzing the different molecules of the aromatic rings, by using different metabolic pathways. The degradation is a complex process, since it involves different cascade enzymatic reactions succeeding each other and requires the participation of more genes. Hydroxylation and oxidation are the main mechanisms leading the enzymatic degradation/transformation of PAHs [18,27]. However, this process is effective with satisfactory results when it is used with LMW-PAHs. Indeed, as the number of rings increases, the number of potential hydroxylation sites by the enzymes of hydroxylation of the rings is also augmented [20,25]. Figure 1 summarizes the complete and integrated degradation of pyrene with the different key enzymes and involved functional genes. The metabolic reactions of pyrene are conditioned by the efficiency of production and function of the enzymes involved in the process. In total, 25 main steps are listed from pyrene to TCA cycle for pyrene degradation by PYR1. Twenty-seven enzymes were identified by Kim et al. [20], as summarized in Table 1. Overall, 14 of them are responsible for the metabolism of pyrene to phthalate, 6 are involved in the transformation of phthalate into protocatechuate, and 7 are implicated in the conversion of protocatechuate to Acetyl CoA and Succinyl CoA. In our previous study [28], by using the bacterial strain *Rhodococcus* sp. P14, we identified in its genome 24 genes encoding for the cytochrome P450 monooxygenase and 6 genes encoding for the ring-hydroxylating oxygenase. In the second step of pyrene metabolism, which consisting of transforming the cleaved compounds to intermediate products such as protochatechuate, genes encoding hydrolase (4), alcohol dehydrogenase (16), carboxymuconolactone decarboxylase (9), acetaldehyde dehydrogenase (4), aldolase (4), and serine hydrolase (4) have been identified in the same genome. Finally, in the third step, which consisting of metabolizing protocatechuate, two genes encoding for protocatechuate 3,4-dioxygenase, one encoding for β-carboxy-cis, cis muconate cycloisomerase, one gene encoding for γ-carboxymuconolactone decarboxylase, three genes encoding for β-ketoadipate-CoA-transferase, and one gene encoding β-Ketoadipyl CoA-thiolase have been identified and characterized in the genome of P14 [28].

*M. vanbaalenii* PYR-1 metabolizes pyrene in two paths: the first path consists of deoxygenating pyrene via C1 and C2 leading to the formation of o-methylated derived to the pyrene 1-2-diol, while the second one is related to the deoxygenation at C4 and C5 of the K-region, resulting in the complete degradation. The deoxygenation of pyrene in C4 and C5 is catalyzed by the ring hydroxylating oxygenase α-subunit (A) and β-subunit (B) encoded by *nidAB* [44]. However, the deoxygenation of pyrene in C1 and C2 to form 1-2-dihydroxypyrene is encoded by *nidA3B3* [44]. It is crucial to emphasize here that the *nidA/B* genes serve as biomarkers and are used to detect bacteria degrading pyrene. Catechol o-methyl transferase is encoded by *MT1743* and catalyzes the transformation reaction of 1-hydroxy-2-methyloxypyrene to 1-2 dimethylpyrene [20]. The metabolic gene *PdoA2B2* encodes phenanthrene ring-hydroxylating oxygenase, α-subunit (A2) and β-subunit (B2), which converts 4-phenanthroate to cis 3-4-dihydroxyphenanthrene-4-carboxylate [45]. The catabolic gene *NidD* encodes the enzyme aldehyde dehydrogenase catalyzing the transformation of 2-carboxylbenzaldehyde into phthalate [46]. Phthalate is converted to phthalate 3-4 dihydrodiol by phthalate 3-4 dioxygenase, α-subunit (Aa) and β-subunit (Bb), encoded by the metabolic gene *PhtAaAb* [47]. The reaction of converting phthalate 3-4 dihydrodiol to 3-4 dihydroxy phthalate is catalyzed by phthalate 3-4 dihydrodiol dehydrogenase encoded by *PhtB* [34].

Biochemical Reactions

The degradation route by which pyrene is deoxygenated via C1 and C2 leading to the formation of o-methylated is the least mastered. Thus, investigations which will permit to fully understand this path must be undertaken. However, the second route is more investigated and mastered. Indeed, after dehydrogenation, pyrene is transformed into cis 4-5-dihydroxy-4-5-dihydropyrene, which is subsequently converted to pyrene cis 4-5 dihydrodiol. This later is rearomatizated and subsequent ring cleavage dioxygenation resulting in the formation of 4-5 dicarboxyphenanthrene, which is then decarboxylated to form 4-phenathroate. A second dioxygenation reaction leads to the formation of subsequent intermediate cis 3-4-dihydroxyphenanthrene-4-carboxylate, which by rearomatization forms 3-4 dihydroxyphenanthrene that, in return, is metabolized to 1-hydroxy-2-naphthoate. The enzymatic reactions involving intradiol ring cleavage dioxygenase occurs in 1-hydroxy-2-naphthoate, resulting in the formation of o-phthalate. In the last step, phthalate is transformed to protocatechuate via the β-ketoadipate pathway to produce succinyl and acetyl coenzyme A (Succinyl CoA and Acetyl CoA), which are engaged in the TCA cycle intermediate, to being totally metabolized to carbon dioxide and ATP [20,25,47,48]. Based on the KEGG database, the complete conversion of citrate to oxaloacetate in TCA cycle was constructed by Kanehisa et al. [49]. Based on this later study, our previous study identified 18 genes encoding the various enzymes involved in the TCA cycle: 3 genes encoding citrate synthase which convert oxaloacetate to citrate; 1 gene encoding aconitate hydratase involving in the conversion of citrate to isocitrate; 1 gene encoding isocitrate dehydrogenase to transform isocitrate to 2-oxoglutarate; 2 genes encoding oxoglutarate ferrodoxinored oxiductase, which is involved in the transformation of 2-oxoglutarate to succinyl-CoA; 2 genes encoding the succinyl-CoA synthetase to convert succinyl-CoA to succinate; 6 genes encoding succinate dehydrogenase to transform succinate to fumarate; 2 genes encoding fumarate dehydratase to convert fumarate to malate; 1 gene encoding malate dehydrogenase to convert malate to oxaloacetate have been characterized and identified in the genome of *Rhodococcus* sp. P14 [28].

**Table 1 ijms-22-08202-t001:** Different key enzymes and functional genes involved in the pyrene metabolic pathway identified in different bacterial strains including *M. vanbaalenii* PYR-1 grown in the presence of pyrene.

Organisms	Functional Genes	Identified Function	Enzymatic Step	References
*M. vanbaalenii* PYR-1	*NidA*	Pyrene/phenanthrene ring-hydroxylating oxygenase, α subunit	1	[29]
*M. vanbaalenii* PYR-1	*NidB2*	Pyrene/phenanthrene ring-hydroxylating oxygenase, α- subunit	1	[20]
*Mycobacterium* sp. strain 6PY1	*PdoA1*	Pyrene/phenanthrene ring-hydroxylating oxygenase, β-subunit	1	[30]
*Nocardioides* sp. strain KP7	*PhdE*	Dihydrodioldehydrogenase	2/6/23	[31]
*M. vanbaalenii* PYR-1	*PhdF*	Ring cleavage dioxygenase	3/7	[20]
*Arthrobacter keyseri* 12B	*PhtC*	Decarboxylase	4/15	[32]
*Mycobacterium* sp. strain 6PY1	*PdoA2*	Phenanthrene ring-hydroxylating oxygenase, αsubunit	5	[32]
*Mycobacterium* sp. strain 6PY1	*PdoB2*	Phenanthrene ring-hydroxylating oxygenase, β-subunit	5	[32]
*M. vanbaalenii* PYR-1	*PhdG*	Hydratase-aldolase	8	[20]
*M. vanbaalenii* PYR-1	*NidD*	Aldehyde dehydrogenase	9	[33]
*Nocardioides* sp. strain KP7	*PhdH*	Aldehyde dehydrogenase	9	[31]
*M. vanbaalenii* PYR-1	*PhdI*	1-Hydroxy-2-naphthoate dioxygenase	10	[20]
*Nocardioides* sp. strain KP7	*PhdI*	1-Hydroxy-2-naphthoate dioxygenase	10	[31]
*M. vanbaalenii* PYR-1	*PhdJ*	*trans*-2_-Carboxybenzalpyruvate hydratase-aldolase	11	[20]
*Nocardioides* sp. strain KP7	*PhdJ*	*trans*-2_-Carboxybenzalpyruvate hydratase-aldolase	11	[31]
*Nocardioides* sp. strain KP7	*PhdK*	2-Carboxylbenzaldehyde dehydrogenase	12	[31]
*M. vanbaalenii* PYR-1	*PhtAa*	Phthalate 3,4-dioxygenase, α subunit	13	[34]
*Terrabacter* sp. strain DBF63	*PhtA1*	Phthalate 3,4-dioxygenase, α subunit	13	[35]
*M. vanbaalenii* PYR-1	*PhtAb*	Phthalate 3,4-dioxygenase, β subunit	13	[34]
*Terrabacter* sp. strain DBF63	*PhtA2*	Phthalate 3,4-dioxygenase, β subunit	13	[35]
*M. vanbaalenii* PYR-1	*PhtAc*	Oxygenase ferredoxin component	1/5/13/22	[36]
*Arthrobacter keyseri* 12B	*PhtAc*	Oxygenase ferredoxin component	1/5/13/22	[37]
*M. vanbaalenii* PYR-1	*PhtAd*	Oxygenase reductase component	1/5/13/22	[36]
*Arthrobacter keyseri* 12B	*PhtAd*	Oxygenase reductase component	1/5/13/22	[37]
*M. vanbaalenii* PYR-1	*PhtB*	Phthalate 3,4-dihydrodiol dehydrogenase	14	[34]
*Terrabacter* sp. strain DBF63	*PhtB*	Phthalate 3,4-dihydrodiol dehydrogenase	14	[35]
*Streptomyces* sp. strain 2065	*PcaG*	Protocatechuate 3,4-dioxygenase, α subunit	16	[38]
*Streptomyces* sp. strain 2065	*PcaH*	Protocatechuate 3,4-dioxygenase, β-subunit	16	[38]
*Terrabacter* sp. strain DBF63	*PcaB*	β-Carboxy-*cis*,*cis*-muconatecycloisomerase	17	[39]
*Rhodococcus opacus* 1CP	*PcaL*	γ-Carboxymuconolactonedecarboxylase/β-ketoadipate enol-lactone hydrolase	18/19	[40]
*Pseudomonas putida* PRS2000	*PcaI*	β-Ketoadipatesuccinyl-CoA transferase, α subunit	20	[41]
*Pseudomonas putida* PRS2000	*PcaJ*	β-Ketoadipatesuccinyl-CoA transferase, β subunit	20	[41]
*Terrabacter* sp. strain DBF63	*PcaF*	β-Ketoadipyl-CoA thiolase	21	[39]
*M. vanbaalenii* PYR-1	*NidA3*	Fluoranthene/pyrene ring-hydroxylating oxygenase, α subunit	1/22	[20]
*M. vanbaalenii* PYR-1	*NidB3*	Fluoranthene/pyrene ring-hydroxylating oxygenase, β subunit	1/22	[20]
*Mycobacterium tuberculosis* CDC1551	*MT1743*	Catechol *O*-methyltransferase	24/25	[42]
*M. vanbaalenii* PYR-1	*MT1743*	Catechol *O*-methyltransferase	24/25	[20]

##### Degradation by *Mycobacterium* sp. KMS

As PYR-1, the KMS strain is one of the bacteria capable of completely metabolizing pyrene into CO_2_. Unlike PYR-1, in the KMS strain, the pyrene-4-5-dione resulting from the cis 4,5-pyrene dihydrodiol transformation is considered as an intermediate product [50]. In fact, once the molecular cis 4-5-pyrene dihydrodiol is formed, a rearomatization occurs to form 4-5 dihydroxypyrene. An oxidation of 4-5 dihydroxypyrene forms pyrene-4-5-dione (a quinone), and, inversely, pyrene-4-5-dione under the action of the pyrene quinone reductase (PQR) enzyme can be reduced to 4-5 dihydroxypyrene. The presence of pyrene-4,5-dione reveals two majors intermediates: phenanthrene-4,5-dicarboxylate acid and 4-phenanthroic acid [51]. These intermediates help, among other, to better understand the ortho-cleavage of the central ring of pyrene identified, to cleave 4,5-dihydroxypyrene.

As in many other strains, in the *Mycobacterium* sp. KMS strain, mono/dioxygenases are the first enzymes involved in the first step of the pyrene degradation. As a reminder, dioxygenase enzymes are multiple protein compounds composed of an electron transport chain and a terminal dioxygenase, in turn composed of more alpha units and fewer beta units. The alpha (α) subunit is the catalytic component and contains two conserved regions, i.e. [Fe2-S2] Rieske center and mononuclear iron binding domain, permitting the transfer of electron to dioxygen, which constitutes the final acceptor [52,53]. Dioxygenase and dihydrodiol dehydrogenase are the second enzymes involved in transforming pyrene 4,5-dihydrodiol into 4,5-dihydroxypyrene. Later in the process, aldehyde dehydrogenase converts 1-hydroxy-2-naphthaldehyde to 1-hydroxy-2-naphthoic acid and 2-carboxybenzaldehyde to o-phthalic acid. In the last step of degradation, the enzymes aldehyde dehydrogenase, hydrolase-aldolase, and phthalate dihydrodiol dehydrogenase are involved. While PRY-1 metabolized pyrene in two different paths, in the KMS strain, it was reported that monooxygenase enzyme can catalyze the initial oxidation of pyrene to pyrene oxide. Indeed, it would not be a surprise to think that monooxygenase enzyme oxide directly catalyzes pyrene to 1-hydroxypyrene, and then to trans 4,5-dihydroxy-4,5-dihydropyrene. However, in both KMS and *vanbaalenii* PYR-1 culture, pyrene oxide was identified as an intermediate product [54,55]. This suggests that there was an intermediate compound between these molecules and which none other than pyrene oxide is. By using the KMS strain, Liang et al. [43] demonstrated that the oxidation of pyrene by pyrene 4-5 monooxygenase results in the formation of pyrene oxide, which is subsequently converted to *trans* 4,5-dihydroxy-4,5-dihydropyrene by epoxide hydrolase. The enzyme dihydrodiol dehydrogenase transforms *trans* 4,5-dihydroxy-4,5-dihydropyrene to 4,5-dihydroxypyrene, which is metabolized by following the same route as in PYR-1, as shown in Figure 1

#### 2.2.2. Anaerobic Degradation

In many natural habitats, such as groundwater, topsoil, marine sludge, and sediments, dioxygen (O_2_) is reduced or even almost absent. Under such anoxic conditions, the metabolism of PAHs in general and pyrene in particular is questionable. However, it has been found that thermodynamically, anoxic mineralization of pyrene in various reducing environments is achievable, despite its lower energy efficiency than aerobic degradation [55]. Although this process is far from being mastered, to date, the mineralization of some PAHs in sulfated, nitrated, iron reducing, and methanogenic environments has been reported as possible by facultative and strict anaerobic bacterial strains and archaea [56,57]. The strain *Clostridium* sp. ER9, a strict rod-shaped Gram-positive bacterium, anaerobically metabolized pyrene up to 93% within 49 days of incubation [58]. In another study, the strain *Paracoccus denitrificans* M-1, a facultative anaerobic bacterium, showed a capacity to degrade pyrene under anoxic conditions with different nitrate/nitrite (NO_3_^−^/NO_2_^−^) ratios within 5–6 days of incubation [59]. Yan et al. [60] reported that the strain *Hydrogenophaga* sp. PYR1 could anaerobically remove pyrene under iron reduction conditions with an efficiency of 22%.

In the aerobic degradation process, oxygen is the final electron acceptor, and, in parallel, it constitutes a co-substrate for the hydroxylation and oxygenolytic cleavage of aromatics rings. However, in anaerobic degradation, nitrogen, metal ions such as ferric, and manganese, sulfide, chlorate, trimethylamine oxide, perchlorate, carbon dioxide, and fumarate instead of oxygen constitute the final electron acceptors. In the anoxic environment, the degradation process of organic compound is challenged by the stability of carbon-carbon (C-C) and carbon-hydrogen (C-H) bonds in the aromatic rings. This is explain why the activation of the compounds in the anoxic medium can occur, by carboxylation, methylation, or the addition of fumarate. During the pyrene mineralization process, with a succession of decarboxylation and hydroxylation, phenanthrene is formed. The mechanism of degradation of phenanthrene obtained from pyrene is anaerobic, consisting of hydroxylation and methylation reactions in the first ring and thus continuing in the other aromatic rings until all the constituent rings of the compound are fully cleaved. Metabolites observed during the degradation of pyrene include 1,2,3,4-tetrahydro-4-methyl-4-phenanthrol, phenol, 2-methyl-5-hydroxy-benzaldehyde, p-cresol, 1-propenyl-benzene, anthraquinone, and 1-anthraquinone-carboxylix acid [61,62,63,64].

Diverse paths are proposed by various studies using strict and optional anaerobic microorganisms metabolizing low and high PAHs. In the following, two facultative different bacterial strains, *Pseudomonas* sp. JP1 and *Klebsiella* sp. LZ6, are used as models to describe the anaerobic degradation mechanism pathway of pyrene.

##### Anaerobic Pyrene Degradation Mechanism by *Klebsiella* sp. LZ6 and *Pseudomonas* sp. JP1

Although the *Klebsiella* sp. LZ6 strain was reported to degrade not more than 33% of 50 mg/L of pyrene, different metabolites were observed and have permitted to propose a degradation pathway for this strain (Figure 2B) [61]. Firstly, pyrene was reduced to pyrene 4,5-dihydro by adding dihydrogen molecules. In this formed compound, carbon-carbon bond cleavage at saturated carbon occurred to form phenanthrene, which subsequently produced 4-hydroxycinnamate, 2-hydroxypropiophenone, and allyl tert-butyl dimethyl phthalate by opening and partially removing the aromatic ring, by a succession of hybridization and hydrolysis reactions. 4-hydroxycinnamate is further converted to p-cresol, by hydrolysis reaction, and then to phenol produced from 4-hydroxybenzyl alcohol, through consecutive oxidation of methyl groups and decarboxylation. 2-hydroxypropiophenone and allyl tert-butyl dimethyl phthalate were subsequently converted to phthalic acid by enzymatic reaction.

4,5-dimethylphenathrene, 4-methylphenanthrene, and phenanthrene dimethyl are the metabolites obtained during the mineralization of pyrene by *Pseudomonas* sp. JP1 (Figure 2A) [62]. The authors stated that the biomineralization of pyrene was initiated by activation and cleavage of the fourth ring at carbon 9 (C9) and carbon 10 (C10) of pyrene, which then produced phenanthrene through a multitude of reactions (Figure 2). Phenanthrene is subsequently converted to 1,2,3,4-tetrahydroxy-4-methyl-4-phenthrenol, 2-methyl-5-hydroxy-benzaldehyde, p-cresol, phenol, and 1-propenyl-benzene. It has been documented that 1,2,3,4-tetrahydroxy-4-methyl-4-phenthrenol is a metabolite of phenanthrene metabolized in a nitrate-reducing environment. Therefore, it was concluded that hydroxylation and methylation could be the activating mechanisms of the anaerobic degradation of phenanthrene.

## 3. Fungi

Along with bacteria, fungi are some of the most important soil microorganisms, not only due to their abundance and the importance of the ecosystem processes, but also because they play a crucial role in the degradation of organic pollutants [65]. In recent years, fungi, specifically filamentous ones (hyphae-type fungi) or molds, have been used more frequently in studies of the degradation of PAHs, unlike bacteria. These fungi show a very efficient ability to degrade a large number of low and high molecular weight PAHs which could not be degraded by other fungi and bacteria. This is due to the activity of extracellular enzyme systems such as laccase, dioxygenase, and peroxidase [66]. An aromatic hydrocarbon degrading fungal population is evaluated to reach up to 1.2^8^ × 10^3^ to 9.6 × 10^6^ CFU/g. Among them, 150 taxonomically and physiologically diverse white rot-fungi, including 55 species, were investigated for their tolerance to PAHs in general and pyrene in particular [67]. Until a few years ago, studies on the degradation of PAHs involving fungi were focused on the white rot-fungi mainly belonging to the Basidiomycete class or Zygomicetes. However, it is clear that other species belonging to classes such as Micromycetes, Ascomycete, Rozellomycota, Mucoromycota, Mortierellomycota, and Glomeromycota are also capable of metabolizing organic pollutants under suitable environmental conditions [68]. Ravalet et al. [69] reported that Micromycetes were able to metabolize pyrene. Additionally, at the genera level, *Peniophora*, *Penicillium*, *Papulospora*, *Cladosporium*, *Aspergillus*, *Phanerochaete*, *Trichaptum*, *Mycoaciella*, *Phlebia*, *Rhizochaete*, *Dentipellis*, *Phlebiella*, *Heterobasidion*, *Pseudochaete*, *Phillotopsis, Bjerkandera,* and *Ceriporia*, are involved in PAH degradation studies and presented higher ability to degrade both low and high PAHs [67,70].

The particular factor which has aroused scientific curiosity in the white rot-fungi involvement on PAH metabolism is their natural ability to degrade lignin, due to the production of complex extracellular enzymes containing laccase (EC1.10.3.2), lignin peroxidase (EC1.11.1.14), and manganese peroxidase (EC1.11.1.13), leading to the complete degradation of PAHs into CO_2_ [67]. Basically, the ligninolytic system contains three main groups of enzymes: lignin peroxidase (LiP), manganese-dependent peroxidase (MnP), and phenol oxidase (laccase and tyrosinase) and H_2_O_2_-producing enzymes [71]. Indeed, the initial attack of PAHs by lignolytic fungi is assured by two mains enzymes: cytochrome P450 monooxygenase and lignin peroxidase. Cytochrome P450 incorporates an oxygen atom into the PAH molecule to form an arene oxide, which undergoes spontaneous isomerization to form a phenol. The three pathways used by fungi to metabolize PAHs are summarized in Appendix A. Zhang et al. [72] reported that the initial oxidation of PAHs is catalyzed by extracellular peroxidases. Both lignin peroxidase and manganese are involved in the other oxidation processes of the metabolism. Peroxidases are directly involved in the oxidation, while manganese peroxidases are indirectly involved in co-oxidation by enzyme-mediated lignin peroxidation. It is interesting that the conditions of production and the activities of these enzymes unconditionally depend on environmental parameters such as temperature, pH, salinity, etc. For instance, it was reported that a pH of 8 enhances high biomass of the strain *Coriolopis byrsina* APC5, while, at pH 7, the maximum production of laccase and MnP occurred and, pH 6 permitted the maximum production of LiP. The temperature of 25 °C resulted in the growth of fungi and production of laccase, LiP and MnP, while 15 and 55 °C enhanced the lignolytic activity of the strain, thus leading to degrade 96% of pyrene at 20 mg/L in 18 days [73]. The effects of pH and temperature on biomass production of *Polyporus* sp. S133 and the production and activity of laccase and 1,2-dioxygenase during the degradation of pyrene were studied by Hadibarata et al. [74]. The optimal values of pH were 3, 5 and 4 for laccase, 1,2-dioxygenase, and biomass production, respectively, while those for temperature were 25 °C for laccase and 50 °C for 1,2-dioxygenase and biomass production.

### 3.1. Fungal Strain with Ability to Metabolize Pyrene

Table 2 presents some strains of fungi with the ability to degrade pyrene. *Trichoderma harzianum, Penicillum simplicissiimum, P. janthinellum, P. funiculosum,* and *P. terrestrial* are able to use pyrene as the sole carbon and energy source. It was reported that, when the concentration of pyrene in the test medium was 50 mg/L, 65% of pyrene was degraded by *T. harzianum*, while, at 100 mg/L, 33% was metabolized. Among the five species, *P. terrestrial* showed the best results in pyrene degradation, by metabolizing 75% at 50 mg/L and 67% at 100 mg/L in 28 days. However, *P. janthinellum* exhibited the lowest performance with only 57% of degradation rate at 50 mg/L and 31.5% at 100 mg/L [75]. *Pseudotrametes gibbasa* was reported to degrade 28.33% of pyrene (10 mg/L) in 18 days [76]. In 30 days, the strain *Armillaria* sp. FO22 degraded 63% of pyrene (5 mg/L) [77]. Khudhair et al. [78] stated that 42% of pyrene at 20 mg/L was metabolized by the strain *Rhizoctonia zeae* SOL3. Overall, 64% of pyrene at 100 mg/L was degraded by *Scopulariopsis brevicaulis* in 30 days [79]. The strain *Pleurotus pulmonarius* FO43 degraded 99% at 10 mg/L of pyrene in 30 days of incubation [80]. The Basidiomycete strain *Marasmiellus* sp. CBMAI 1062 has been reported to degrade almost 100% at 0.08 mg/mL of pyrene in just 48 h under saline condition [81]. The *Peniophora incarnate* strain KUC8836 after two weeks of incubation recorded a higher degradation rate for phenanthrene (95.3%), fluoranthene (95.0%), and pyrene (97.9%). This high performance was explained by the capacity of this strain to produce laccase, LiP, and MnP in higher quantities [67]. The strain *Phlebia brevispora* KUC9045 was used to investigate the degradation of phenanthrene, anthracene, fluoranthene, and pyrene and the production of extracellular laccase and MnP during degradation process. This strain was effective in metabolizing the four PAHs with a degradation rate of 66.3%, 67.4%, 61.6%, and 63.3%, respectively, for phenanthrene, anthracene, fluoranthene, and pyrene [82]. Similarly, *Peniophora incarnata* KUC8836 and *Mycoaciella bispora* KUC8201 were reported to have significant rates of pyrene removal (82.6%) after 2 weeks of incubation [83].

**Table 2 ijms-22-08202-t002:** Pyrene-degradation efficiency of some different fungal strains.

Strains Names	Strains/Taxonomic ID	Degradation Rates (%)	References
*Armillaria* sp.	FO22	63	[76]
*Aspergillus ficuum*	MB#5058	54.6	[84]
*Aspergillus flavus*	MB#347788	59.8	[84]
*Aspergillus fumigatus*	MB#352615	59.6	[84]
*Cladosporium* sp.	CBMAI 1237	62	[85]
*Coriolopis byrsina*	APC5	96.1	[73]
*Coriolopsis byrsina*	APC5	96.1	[86]
*Crinipellis campanella* *Crinipellis perniciosa* *Crinipellis stipitaria*	MB#285848 MB#500896 MB#100767	39 95 94	[87] [87]
*Fusarium* sp.	FJ613115.1	18.2–74.6	[88]
*Marasmiellus* sp.	CBMAI 1062	98.8	[81]
*Marasmiellus ramealis* *Marasmius rotula*	MB#71897 MB#156778	76.5 95	[87]
*Merulius tremellosus*	KUC9161	83.6	[89]
*Penicillum janthinellum*		31.5 and 57	[75]
*Penicillum terrestre*		67 and 75	[75]
*Peniophora incarnata*	KUC8836	82.6	[83]
*Peniophora incarnata*	KUC8836	97.9	[67]
*Phanerochaete chrysosporium*		92.2	[90]
*Phlebia brevispora*	KUC9045	63.3	[82]
*Pleurotus pulmonarius*	FO43	99	[80]
*Polyporus* sp.	S133	71	[74]
*Pseudotrametes gibbasa*		28.33	[76]
*Rhizoctonia zeae*	SOL3	42	[78]
*Scopulariopsis brevicaulis*	PZ-4	64	[79]
*Trichoderma harzianum*		33 and 65	[75]
*Trichoderma* sp.	MB#512453	37.4	[91]
*Trichoderma* sp.	F03	78	[66]

A species tolerance to stress in general, and pollutants in particular, is reflected in its ability to survive and grow under changing conditions. Microorganisms get their nutrients from the environment where they live. Although they are supposed to be toxic, organic pollutants can serve as a source of carbon and energy for organisms which present the required genetic characteristics and adaptation capacity. The survival of organisms among so many others is sufficient proof that they present criteria to wonder if they are capable of using these pollutants as the only source of survival and opens the subject to the optimum conditions for the metabolism of PAHs. The strains *Agrocybe dura* FBCC478, *Agrocybe praecox* FBCC587, *Gymnopilus luteofolius* FBCC466, *Irpex lacteus* FBCC1012, *Mycena galericulata* FBCC598, *Phanerochaete velutina* FBCC941, *Stropharia aeruginosa* FBCC521, *Stropharia rugosoannulata* FBCC475, and *Trametes ochracea* FBCC1011 showed significant growth on plates with a combination of 16 PAHs [92]. Similarly, in three months of incubation, the strain *Phanerochaete velutina* showed a degradation rate of 96% and 39%, respectively, for PAHs with 4, 5, and 6 rings [93]. On the other hand, they reported that the strains *Rhizochaete* sp. KUC8364 and *Dentipellis dissita*. KUC8613 developed perfectly well in the media rich in fluoranthene and pyrene, with a growth rate ranging 90–100%, while that of the strains *Phanerochate calotricha* KUC8040 and KUC8003 in the same media ranged 70–90% [67,82,83].

### 3.2. Degradation of Pyrene by Fungi

The ability of aerobic fungi to break down PAHs relies on oxygenase enzymes which reduce elemental oxygen in order to activate the fungi, which then led them to enter the substrates. The aerobic oxidation of pyrene by fungi starts with the configuration of dihydrodiol by an enzymatic cascade of dioxygenases, which is later used either by the meta or ortho pathway, resulting in the formation of intermediates such as catechol and protocatechuate [93]. Aerobic conditions occur in electron acceptors with low standard reduction potential and convert them to smaller standard Gibbs free energy. It is fundamental to point out here that different fungi involved in the mineralization of PAHs in general and pyrene in particular are divided into lignolytic and non-lignolytic fungi. The initial attack of PAHs by lignolytic fungi is insured by two main enzymes: cytochrome P450 monooxygenases and lignin peroxidase. However, for the non-lignolytic fungi, the oxidation by monooxygenase enzymes such as cytochrome P450 monooxygenase and free dioxygen (O_2_) in order to create arene oxide and water is the initial transformation of PAHs in general and pyrene in particular. It is crucial here to notify that both lignolytic and non-lignolytic fungi such as *Aspergillus, Penicilium, Phanaerochate, Pleurotus, Tramates,* and *Cunninghamella* are reported to have the ability to produce cytochrome P450 monooxygenase during the degradation of PAHs having 2-5 aromatic rings [94] The lignolytic fungi *P. ostrestus* can metabolize several PAHs including pyrene. The extracellular enzymes laccase, LiP, and MnP oxidize organic compounds and create PAH transient diphenols, which are automatically oxidized to quinone thereafter [95,96]. The laccase enzyme oxidizes PAHs in the presence of mediator compounds such as aniline, 4-hydroxybenzoic acid, phenol, methionine, cysteine, 4-hydroxyl benzyl alcohol, or reduced glutathione [97,98]. The point to be noted here is that the laccase/mediator system does not require direct contact between the oxide compound and the enzyme. In fact, the mediator is initially oxidized by laccase and the radical produced (radical cation or free radical) in turn leads to the oxidation of PAHs to quinone, polymers or high molecular weight, and a small amount of CO_2_ [98,99] (Appendix A).

#### 3.2.1. Aerobic Pyrene Degradation and *Coriolopis byrsina* APC5

The mechanism pathway of aerobic pyrene degradation by *Coriolopis byrsina* APC5 strain is summarized in Figure 2C as follows: pyrene is oxidized by the action of laccase in the 4-5 pond (K-region) to pyrene oxide, which in turn is converted to trans 4,5-pyrene dihydrodiol. The transformation by dehydrogenation of *trans* 4,5-pyrene dihydrodiol resulted to 4,5-dihydroxypyrene. The later formed compound, in an oxidation reaction by laccase, LiP, and MnP and ring cleavage process, is converted to phenanthrene, which is subsequently metabolized to phthalic acid diiosopropyl ester and then to benzoic acid under a succession of enzymatic reactions involving the extracellular enzymes laccase, LiP, and MnP. From the benzoic acid, the side chain is removed and by ring cleavage and, under the action of laccase, LiP, and MnP, benzoic acid is converted to pyruvic acid, which is subsequently metabolized in the metabolic pathway to carbon dioxide [73].

#### 3.2.2. Anaerobic Degradation

It is conventional that the main obstacle to the microbial degradation of PAHs is the stabilizing resonance energy of aromatic compounds. However, in addition to that comes the unavailability of oxygen in an anaerobic environment, presenting another critical challenge for microorganisms. The degradation of PAHs in general and pyrene in particular in an oxygen-deprived medium occurs in several phases depending on the strains involved. Indeed, generally in these media, pyrene plays the role of the electron donor substrate, and, in the presence of an electron terminal acceptor, namely several inorganics ions or compounds such as acetate, lactate, pyruvate, sodium chloride, cellulose, or zero-valent iron, the microorganisms benefit from the oxidation of these compounds for their growth. The anaerobic metabolism processes of pyrene, similar to those of other aromatic compounds, require less energy and strict temperature control. Anaerobic fungi are organisms with an alternating lifecycle between zoospores, which are flagella and active forms and thallus, which unlike zoospore are vegetative and non-active productive forms. While numerous strains are reported to have the ability to anaerobically degrade pyrene, their mechanism pathway in the anoxic condition are rare and confused. Thus, in Appendix A, we present a general, simplified representation of anaerobic PAH degradation by fungi instead of a specific degradation of pyrene. In the absence of oxygen, diverse radicals or inorganic compounds serve as terminal electron acceptor. The first step of PAH biodegradation, as for the aerobic process, is the activation. For instance, glycyl radical enzyme catalyzes the reaction between PAHs and fumarate to produce aromatic succinates, which subsequently, by methylation and hydroxylation, produce intermediate products. These products formed from methylation and carboxylation are directly engaged in the carboxylation cycle, resulting in unsaturated rings that are subsequently transformed by beta oxidation, producing carbon dioxide, water, and other metabolites [93].

## 4. Degradation of Pyrene by a Consortium

Bioremediation involving bacterial strains, fungi, or microalgae capable of degrading organic compounds is a globally accepted reality. Several strains of microorganisms have been shown to be effective in the bioremediation of organic pollutants and many have been extensively studied to establish the different mechanism pathways for certain compounds such as pyrene. Strains have individually shown impressive efficacy; however, it is obvious and more interesting that, when two or more microbial populations (bacterium–fungus, bacterium–microalga, or microalga–fungus) are in consortium, the result would be more significant compared to when they are involved individually. Liu et al. [100] studied the synergistic degradation of pyrene by a co-culture of bacteria (*Bacillus* sp. and *Sphingomonas* sp.) with a strain of fungi (*Fusarium* sp.). These microorganisms are reported each having the ability to use pyrene as a carbon source and energy. However, in their consortium, the authors showed that the initial concentration of pyrene (100 mg/kg) was degraded by more than 96% in the liquid medium and 87.2% in the contaminated soil. The synergy between different strains in the metabolism of PAHs is explained in the sense that each of these different strains breaks down PAHs. For example, in a bacteria–fungi consortium, it is obvious to imagine that the fungi initiate the first stage of oxidations and then the bacteria perform other oxidations, as shown in [101]. In a consortium between bacteria-microalgae (*Rhodococcus wratislaviensi* strain 9 and *Chlorella* sp. strain MM3), the bacteria strain is used as a heterotrophic partner and that of the microalgae as a phototrophic partner. The bacterial strain *Mycobacterium* sp. B2 individually was effective in the degradation of pyrene up to 82.2% and 83.2% for free and immobilized cells after 30 days of incubation. Thus, although the *Mucor* sp. F2 strain was not as efficient as the B2 strain, their consortium was much more efficient. The degradation rate in the consortium was increased by 51.6% compared to the cumulative rate of pyrene degraded by each individually in an alkaline medium [102]. In a consortium between microalgae and bacteria, the advantages are as enormous as the individual culture of each. Microalgae can, for example, provide oxygen and carbon, two key elements in the aerobic degradation of PAHs. In return, the CO_2_ released by bacteria during respiration can be used by microalgae, without of course omitting the important role that the surface of microalgae can play as a stable habitat for bacteria. Consequently, in a perspective that each strain can individually degrade PAHs, their mixture will have more interesting results compared to their individual effect. Luo et al. [103] realized a consortium with a microalga (*Selenastrum capicornutum*) and a bacterium (*Mycobacterium* sp. strain A1-PYR) to evaluate the degradation of pyrene. The result reveal that the joint and mutualistic effect of the consortium was significant, not only in the degradation of pyrene but also in the growth of bacteria [103].

## 5. Conclusions and Future Perspectives

A better understanding of bioremediation requires a great mastery of interactions and important ecological relationships between different microorganisms. As astonishing as it may seem, it is not easy to study dynamic microbial interactions in real time because of the numerous uncultivable microorganisms and limited molecular tools. However, as pointed out by Professor Gu, in a microbial culture, a target pollutant under investigation is not the only source of carbon and energy for the development of microorganisms. Such results would be the work of a biochemical process called co-metabolism. In the present case, co-metabolism is an efficient and common route for the natural biodegradation of pyrene, whether in a monoculture or a microbial consortium. Technology and science have increasingly developed powerful tools such as multi-omics techniques including metagenomics, transcriptomics, and proteomics for the reconstruction of interactive networks of microorganisms in purpose of PAHs biodegradation. However, for a true construction of a mechanism and degradation pathway, the culture medium must be controlled so that the constituent compounds do not interfere with the degradation process. For this, a defined mineral medium with a chemical composition known to achieve the objective of degradation by microorganisms capable of using the test chemical as the sole source of carbon and energy is the key motor. Therefore, using the appropriate technological means such as RT-qPCR, microarray, PCR-based fingerprints, and biosensors, the different actors of degradation including catabolic genes, enzymes, and intermediates can be determined and identified. Besides, to understand the medium for a real reconstruction of pathway degradation, future studies on the anaerobic degradation of pyrene with strains already known to have degradation capacity would be a unique initiative to propose a complete and integrated pathway on the mechanism of degradation of PAHs in general and pyrene in particular. It is evident that anaerobic fungi have shown some potential for the metabolism of organic pollutants including PAHs. It is therefore recommended to use the combination of various strains of anaerobic fungi and anaerobic fungi–bacteria consortium to achieve maximum efficiency of the biodegradation mechanism of pyrene.

## Figures and Tables

**Figure 1 ijms-22-08202-f001:**
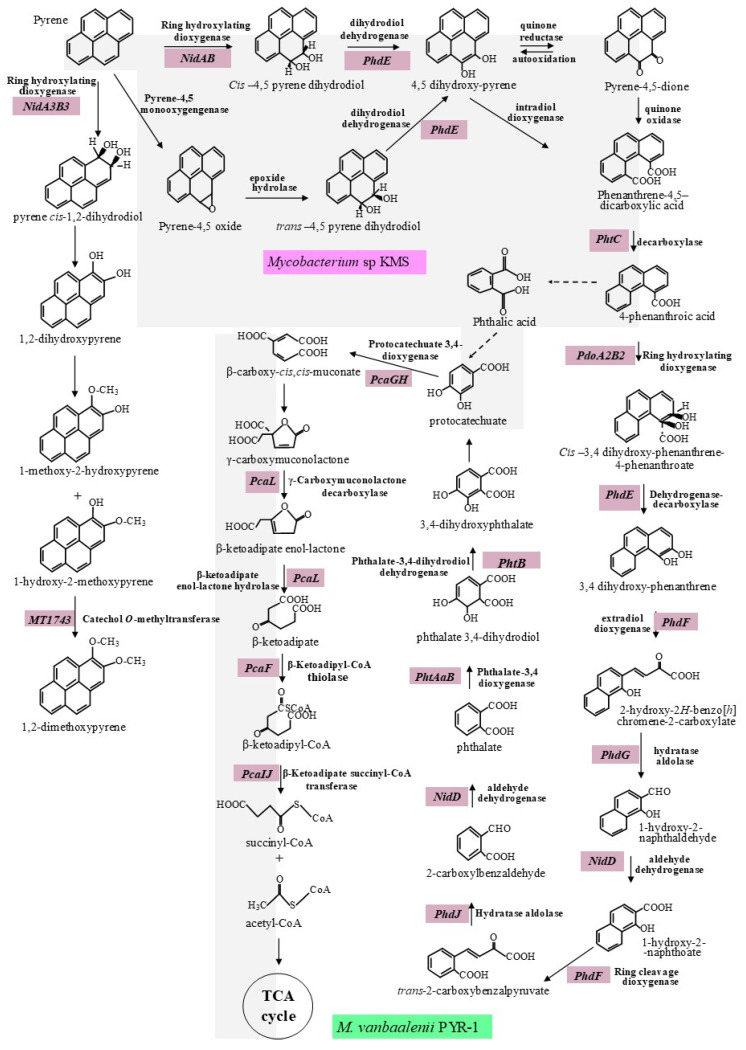
Complete aerobic pyrene degradation pathway by *M. vanbaalenii* PYR-1 (no color side) and *Mycobacterium* sp. KMS (the colored one). The half colored steps are shared by the two strains. Solid and dashes arrows represent single and multiple steps, respectively (adapted from Liang et al. [43] and Kim et al. [20]).

**Figure 2 ijms-22-08202-f002:**
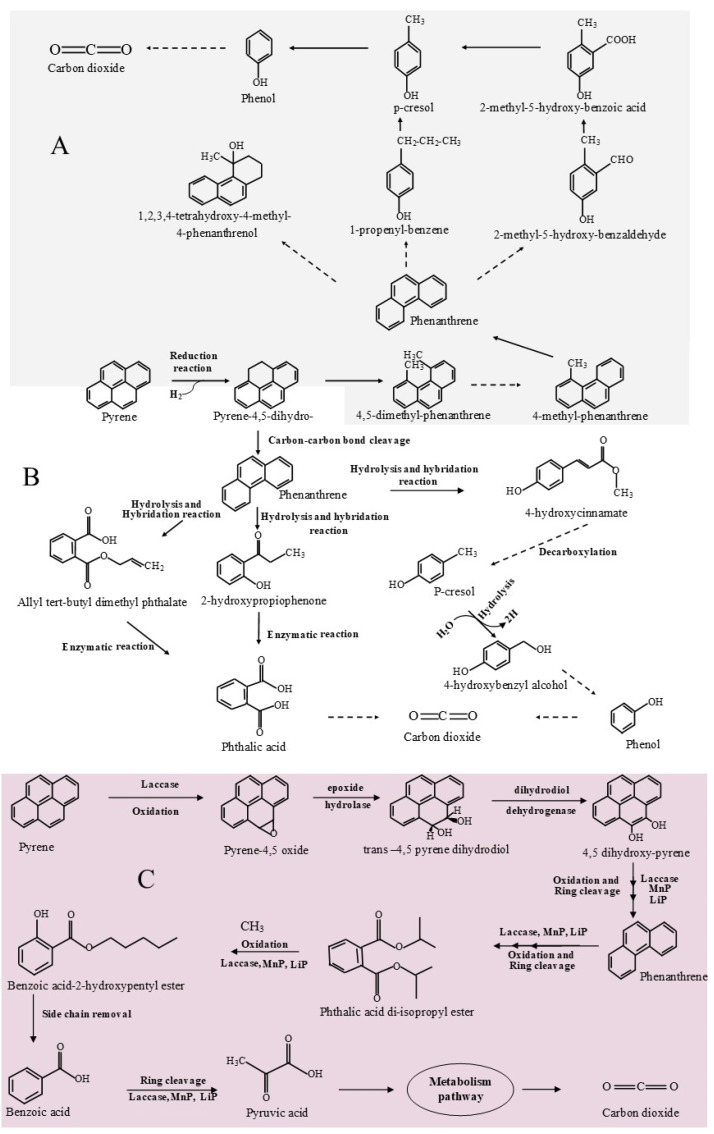
Pyrene anaerobic degradation pathway by *Pseudomonas* sp. JP1 (**A**), *Klebsiella* sp. LZ6 (**B**), and *Coriolopis byrsina* APC5 (**C**). Solid and dashed arrows represent, respectively, single and multiple steps (adapted, respectively, from Liang et al. [62]; Li et al. [61] and Agrawal and Shahi [63]).

## Data Availability

Not applicable.

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
