# Peer review of "Aerobic and Anaerobic Bacterial and Fungal Degradation of Pyrene: Mechanism Pathway Including Biochemical Reaction and Catabolic Genes"

_ijms, 2021, doi:10.3390/ijms22158202_

Round 1

Reviewer 1 Report

The manuscript “Aerobic and Anaerobic Bacterial and Fungal degradation of Pyrene: Mechanism pathway including Biochemical reaction and Catabolic Genes” was well written and well structured. But there is a scope of improvement. I found that, the principal biomarkers of pyrene metabolism (nidA/B, pah, and phd genes) present in bacterial community which are associated with pyrene biodegradation mechanisms was recently been reviewed, which was also been cited by the author in the submitted manuscript. However, they have not discussed the involvement of fungal community in their review paper, which brings the nobility of the manuscript. I found there were several names of pathogenic microorganisms listed in the manuscript which have been studies against the Pyrene degradation ability, such as, Mycobacterium tuberculosis, Aspergillus flavus and so on. Therefore, I think it will be interesting to discuss the pros and cons associated with the usage of these microorganism to reduce pyrene as an environmental pollutant.

Author Response

Thank you for your valuable comments. We have revised this manuscript according to your suggestions. We hope that the revised manuscript would meet your requirements for publication in IJMS.

General comment:

The manuscript “Aerobic and Anaerobic Bacterial and Fungal degradation of Pyrene: Mechanism pathway including Biochemical reaction and Catabolic Genes” was well written and well structured. But there is a scope of improvement. I found that, the principal biomarkers of pyrene metabolism (nidA/B, pah, and phd genes) present in bacterial community which are associated with pyrene biodegradation mechanisms was recently been reviewed, which was also been cited by the author in the submitted manuscript. However, they have not discussed the involvement of fungal community in their review paper, which brings the nobility of the manuscript. I found there were several names of pathogenic microorganisms listed in the manuscript which have been studies against the Pyrene degradation ability, such as, Mycobacterium tuberculosis, Aspergillus flavus and so on. Therefore, I think it will be interesting to discuss the pros and cons associated with the usage of these microorganism to reduce pyrene as an environmental pollutant.

Response:

First of all, we thank you for all the raised suggestions and comments. As you mentioned that the principal biomarkers of pyrene metabolism (nidA/B, pah, and phd genes) present in bacterial community which are associated with pyrene biodegradation mechanisms was recently been reviewed. In the present manuscript the biomarkers was just cited and not thoroughly discussed, since the recent published review that you have mentioned not only has already reviewed that section, but also we are in the same laboratory and belonging to the same research group, so to avoid repetition, we just briefly quote that section. However as you have noticed, in that review the fungi community was not included, this is why we have discussed this section in the present manuscript.

The reviewer raised a crucial point, when he talk about the use of certain pathogenic microorganisms in bioremediation. This is a much more interesting subject in the context of pros and cons comparative studies and therefore we confess that it would have given a whole new value to this present manuscript. However, if we have not incorporated the discussion on the pros and cons of pathogenic microorganisms that are involved in bioremediation, it is just to keep the aim and objective of the manuscript which is to provide key information on the various steps and actors involved in the bacterial and fungal aerobic and anaerobic degradation of pyrene, by bacteria and fungi, including catabolic genes and enzymes. Furthermore, as you have probably noticed, we just mentioned these microorganisms on the list of strains with the ability to metabolize pyrene. We are not the ones who performed the experience involved these pathogenic microorganisms. This present is a review which deals with the biochemical mechanism of microorganisms and the molecular and biochemical actors involved in the pathway degradation of pyrene. If bacteria like Mycobacterium tuberculosis is listed here, it is to defer its biochemical role in producing an enzyme involved in the metabolism of pyrene. It is obvious that if in a strain, gene X having Y function is identified, if this same gene is also identified on another strain, whether it is virulent or not, the metabolic and biochemical function of the gene can be exploited. We assumed that this could be the goal sought by the author who used the fungi pathogens such as Aspergillus flavus and so on, for bioremediation purpose. However, we are completely in agreement with you, that the use of microorganisms in bioremediation should be done following a comparative study of the advantages and disadvantages that can result from its use in public health and the ecosystems concerned.

Finally, we thank the reviewer for this relevant remark, which in the near future, you might see us with a comparative study including pros and cons of microorganisms with adverse effects and which are involved in the bioremediation of organic pollutants.

Reviewer 2 Report

The publication is well written and includes a comprehensive study of knowledge on a given topic. Therefore, my only remark is the style of the language and the need to correct grammar. Besides, I accept the work as it stands.

Author Response

Thank you for your valuable comments and suggestions. We are quite sure that without you valuable comments and suggestions this work will not have the value that it had to. So thank you

General comment

The publication is well written and includes a comprehensive study of knowledge on a given topic. Therefore, my only remark is the style of the language and the need to correct grammar. Besides, I accept the work as it stands

Response: the manuscript was sent to two English native speaker and experts in scientific language editors, to correct and check grammatical error

You can find their contacts below

Dr Akami Mazarin (Nigeria Academic Bureau): [email protected]

Dr Georges Paterson (Science and Education bureau): [email protected]  
